# Comparing long-term value creation after biotech and non-biotech IPOs, 1997–2016

**Ekaterina Galkina Cleary[1], Laura M. McNamee[2], Skyler de Boer[3], Jeremy Holden[3], Liam Fitzgerald[3], Fred D. Ledley[4]\***

**1** Department of Mathematical Sciences, Center for Integration of Science and Industry, Bentley University, Waltham, Massachusetts, United States of America, **2** Department of Natural & Applied Science, Center for Integration of Science and Industry, Bentley University, Waltham, Massachusetts, United States of America, **3** Center for Integration of Science and Industry, Bentley University, Waltham, Massachusetts, United States of America, **4** Department of Natural & Applied Science, Department of Management, Center for Integration of Science and Industry, Bentley University, Waltham, Massachusetts, United States of America

\* fledley@bentley.edu

**Data Availability Statement:** All relevant data are within the manuscript and its Supporting information files.

## Abstract

We compared the financial performance of 319 BIOTECH companies focused on developing therapeutics with IPOs from 1997–2016, to that of paired, non-biotech CONTROL companies with concurrent IPO dates. BIOTECH companies had a distinctly different financial structure with high R&D expense, little revenue, and negative profits (losses), but a similar duration of listing on public markets and frequency of acquisitions. Through 2016, BIOTECH and CONTROL companies had equivalent growth in market cap and shareholder value (> $100 billion), but BIOTECH companies had lower net value creation ($93 billion vs $411 billion). Both cohorts exhibited a high-risk/high reward pattern of return, with the majority losing value, but many achieving growth multiples. While investments in biotechnology are often considered to be distinctively risky, we conclude that value creation by biotech companies after IPO resembles that of non-biotech companies at a similar stage and does not present a disproportionate investment risk.

## Introduction

Since the first Initial Public Offering (IPO) of a modern biotech company by Genentech in 1980, public markets have played a critical role in the financing of the biotechnology industry. From 1980–2016, >750 biotech companies completed IPOs, collectively raising >$200 billion in capital from public markets (https://finance.yahoo.com/news/historical-biotech-ipo-data-godfather-183606131.html). These investments were based on the promise that investments in emerging biotechnologies, from gene cloning and genomics to combinatorial chemistry and bioinformatics, would make pharmaceutical R&D more efficient, bring new classes of products to market, and establish a sector of profitable biotech companies. Today, there is a vibrant biotechnology sector, led by S&P 500 companies such as Amgen, Biogen, Gilead, Incyte, Regeneron, and Vertex. Many other biotech companies have been acquired, sometimes in blockbuster transactions, such as the acquisitions of Celgene by Merck, Genentech by Roche,

**Funding:** This work was supported by a grant from the National Biomedical Research Foundation, a 501(c)(3) non-profit, awarded to FDL. The National Biomedical Research Foundation had no role in the study design, data collection and analysis, decision to publish, or preparation of the manuscript.

**Competing interests:** The authors have declared that no competing interests exist.

Genzyme by Sanofi, and Pharmacyclics by Abbvie. Hundreds of other companies, however, have languished, been consolidated to ensure survival, or ceased operations without achieving commercial success.

There has been little formal research on the long-term outcomes of investments in public biotech companies. In a landmark study, Pisano examined the track record of the industry from its inception through 2003 [1, 2]. His analysis argued that the biotechnology sector had not yet matured beyond its early focus on advancing science and managing intellectual property. He showed that very few biotech companies generated positive cash flow or profit, and, with the exception of Amgen, the entire sector exhibited negative "operating income before depreciation (cash flow)" [2]. Moreover, he observed that many biotech companies were moving away from their distinctive focus on translating cutting edge biomedical science into novel products and were focusing instead on licensing latter stage projects, as well as repurposing or reformulating existing compounds.

McNamee and Ledley have described the ten-year follow-up of 46 biotech companies that completed IPOs during the short IPO window in 2000 [3, 4]. They showed that companies with early-stage, "nascent" technologies had the highest valuations at IPO, but uniformly failed to translate these technologies into a product pipeline and had market caps in 2010 (10 years after IPO) that were lower than at the time of IPO. In contrast, companies with "established" technologies at the time of IPO, generated a robust product pipeline and positive returns through a turbulent economic decade [3, 4].

Both of these studies, and others [5–7], have called attention to the challenge of valuing public companies with an architecture characterized by pre-commercial stage science and technology, little revenue, negative earnings, and largely intangible assets. There is also persistent concern about the high cost and failure rate of products in clinical development [8, 9] and the disproportional negative effect of clinical failures on stock prices (https://www.forbes.com/sites/brucebooth/2011/11/18/risky-business-late-stage-vs-early-stage-biotech/#12c513662cd6) [10, 11]. These concerns have perpetuated the perception that biotechnology represents a uniquely high-risk investment for public investors.

Analysts have often been pessimistic about the role of public financing for biotechnology. This pessimism was particularly acute during the twin recessions of 2001–2002 and 2008–2009 [12], when public markets were largely closed to biotech companies. These downturns, however, were followed by record numbers of biotech IPOs during the windows from 2004–2007 and 2013-present, which had brought unprecedented amounts of new, public capital into the biotechnology sector (https://finance.yahoo.com/news/historical-biotech-ipo-data-godfather-183606131.html).

This study examines the finances and value creation of emerging public biotech companies compared to a paired set of non-biotech companies from 1997–2016. We focus specifically on 319 biotech companies aimed at developing therapeutic products with an IPO from 1997–2016 ("BIOTECH" dataset) in comparison to non-biotech companies with contemporaneous IPO dates ("CONTROL" dataset). This design pairs each biotech company with a non-biotech that faced equivalent market and economic conditions in the years after IPO. To our knowledge, this is the first such comparison of biotech and non-biotech companies after IPO. We describe the length of time companies were listed on public exchanges; reasons for delisting (including frequency of acquisition); annual revenue, R&D expense, and profit; capital raised and returned to shareholders; and three measures of value creation including market cap, shareholder value, and net value creation.

Our analysis shows that, while this cohort of biotech companies exhibits the distinct architecture described by Pisano, with high R&D expense, limited revenue, and negative earnings, they perform as well as paired controls in most measures of economic performance. The broad

similarities in the performance of biotech and non-biotech companies suggests that the high-risk/high-return profile should not be seen as a distinctive feature of the biotechnology sector and its architecture, but rather a characteristic of emerging firms onto public markets.

## Results

### Company characteristics through the IPO

We identified 319 biotech companies (hereafter, "BIOTECH") focused on developing therapeutic products that completed an IPO on NASDAQ from January 1, 1997 to December 31, 2016. The financial performance of these companies was compared to 319 paired, non-biotech companies (hereafter, "CONTROL"), from a wide range of industrial sectors, with contemporaneous IPO dates on NASDAQ. The average interval between the BIOTECH IPO and that of the paired CONTROL was 4.8 days (median 2 days). The paired sample design controls for market conditions in the years following IPO, when many companies remain dependent on capital markets for operating capital and growth. Companies and summary statistics are described in S1 Table and Table 1.

Fig 1 shows the date and market cap at the end of the first day of trading for BIOTECH and CONTROL companies. We distinguished five different windows of IPO activity from 1997–2016: 1997–1998, 1999–2002, 2003–2007, 2009–2012, and 2013–2016. These windows correspond generally to the IPO cycles described by Papodololus and others (https://finance.yahoo.com/news/historical-biotech-ipo-data-godfather-183606131.html), except that we separate the current IPO window into an early window (2009–2012) and late window (2013–2016).

We consider the first market cap to be the closing stock price on the first day of trading. This corresponds to the "fair market value" of the company as well as the "actual offer price" used in the analysis by Guo [13] and eliminates potential bias associated with underpricing of IPOs by underwriters (https://site.warrington.ufl.edu/ritter/files/2018/07/IPOs2017Underpricing.pdf) [5, 14, 15]. While there was extensive overlap in the first market cap of BIOTECH and CONTROL, the market cap of BIOTECH was lower (median $236.8M vs $443.4M, p<0.005).

### Company fates

As of the end of the study period (December 31, 2016), 224 (35%) of companies (34.5% BIOTECH, 35.7% CONTROL) had been delisted and 414 (65%) (65.5% biotech, 64.3% controls)

**Table 1. Summary financial metrics of BIOTECH and paired CONTROL companies from IPO to 2016.**

| | Median (Q1, Q3) ($ Million) | | Mann-Whitney P-value* | Cumulative totals ($ Billion) | |
|---|---|---|---|---|---|
| | Biotech | Controls | | Biotech | Controls |
| IPO price | 12.2 (9.3, 16.7) | 15.7 (11.3, 19.5) | <0.0001 | N/A | N/A |
| First market cap | 237 (147, (465) | 443 (183, 874) | <0.0001 | 122 | 420 |
| Last market cap | 268 (59.8, 716) | 335 (81.2, 846) | >0.5 | 249 | 553 |
| End shareholder value | 283 (67.3, 716) | 397 (102, 968) | 0.3 | 254 | 588 |
| IPO offer size | 68.6 (50.8, 99.8) | 101 (58.4, 169) | <0.0001 | 25.8 | 51.1 |
| Pre-IPO investment | 94.3 (60.4, 133) | 49.0 (10.9, 124) | <0.0001 | 34.9 | 50.6 |
| Post-IPO stock sales | 25.0 (2.0, 78.2) | 4.23 (0.35, 37.0) | <0.0001 | 101 | 75.6 |
| Total capital raised | 392 (256, 635) | 302 (189, 519) | 0.002 | 161 | 176 |
| Net capital raised | 372 (254, 623) | 264 (138, 465) | <0.0001 | 156 | 142 |

Values are in USD, inflation adjusted to 2016; p-values $< 10^{-4}$ are rounded to "0.0001."

*P-values are adjusted for multiple comparisons by applying a Bonferroni correction of n = 10.

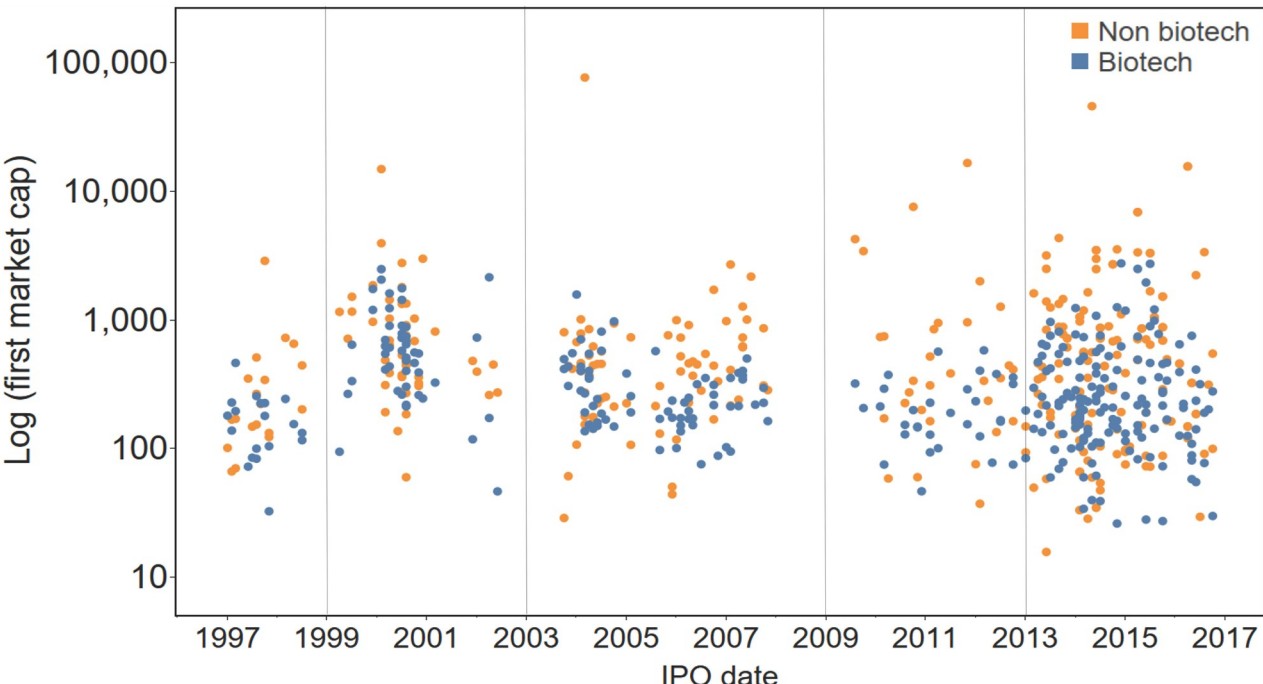

**Fig 1. First market cap and IPO date of BIOTECH (blue) and non-biotech CONTROL (orange) companies showing IPO windows.** For each BIOTECH IPO a paired CONTROL was identified with a concurred IPO date (median difference of 2 days). First market cap is determined at the first day of trading.

were still active. Most of the active companies had IPOs in the most recent 2013–2016 window (n = 274). As shown in Table 2, the fraction of BIOTECH companies delisted was not greater than CONTROL companies. For both BIOTECH and CONTROL, the most common reason for delisting was merger or acquisition (29.5% vs 28.5%). Bankruptcy or liquidation was infrequent for both cohorts (3.1% vs 2.5%).

We performed a Kaplan-Meier time-to-event analysis [16] to calculate the probability of a delisting event and to compare the length of time between IPO and delisting. This method is commonly used in clinical trials to calculate the probability of patient survival and takes into account events that may not yet have occurred before the study ends (right-censored data) [17]. This method treats the disproportionate number of IPOs in the current window whose fate is not yet known as right-censored data.

The median time from IPO to delisting was not different for BIOTECH and CONTROL (9 yrs vs 10 yrs, p = 0.56), and there was no apparent difference in projected 20-year survival (Fig 2A). We examined the time to delisting for BIOTECH (Fig 2B) and CONTROL (Fig 2C)

**Table 2. Fate of BIOTECH and paired CONTROL companies as of the end of the study period (December 31, 2016).**

|  | # Active (%) | Acquired | Bankruptcy/ liquidation | Inactive, other reason* |
|---|---|---|---|---|
| BIOTECH | 209 (65.5%) | 94 (29.5%) | 10 (3.1%) | 6 (1.9%) |
| CONTROL | 205 (64.3%) | 91 (28.5%) | 8 (2.5%) | 15 (4.7%) |

Fate of companies that were delisted before the end of the study period (not active) was identified in Compustat (reason for deletion, DLRSN). Other included: Reverse acquisition (1983 forward); Other (no longer files with SEC among other possible reasons), but pricing continues; Now a private company; Other (no longer files with SEC among other reasons).

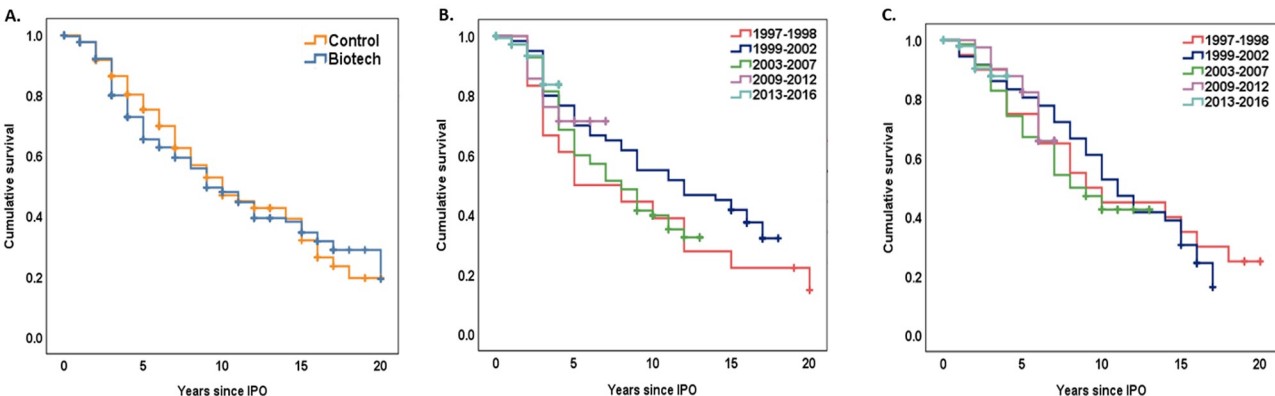

**Fig 2.** A-C. Kaplan-Meier survival analysis of time from IPO to delisting for BIOTECH and paired CONTROL companies or by IPO window. A) Median time to delisting is 9 years for BIOTECH (blue) and 10 years for CONTROL (orange), ($\chi^2$ (1) = 0.35, Log rank p = 0.56), B) BIOTECH only by IPO window ($\chi^2$ (4) = 5.0, Log rank p = 0.29), C) CONTROL only by IPO window ($\chi^2$ (4) = 1.1, Log rank p = 0.9).

for each IPO window. For both BIOTECH and CONTROL, the interval between IPO and delisting was similar for each window (S1 Fig). While there is limited data for companies in the present IPO window, companies appear to be following the trends of earlier windows.

## Financial structure of BIOTECH companies and paired CONTROL companies

We compared the financial performance of biotech companies and controls using several metrics included in audited, annual reports. The first is revenue, also called sales, which reflects the amount received by the company from the sale of products or services. The second is R&D expense. For accounting purposes, research is defined as the cost of "planned search or critical investigation aimed at discovery of new knowledge with the hope that such knowledge will be useful in developing a new product or service, or a new process or technique, or in bringing about a significant improvement to an existing product or process," while development is defined as "the translation of research findings or other knowledge into a plan or design for a new product or process or for a significant improvement to an existing product or process" (https://asc.fasb.org/glossarysection&trid=2127277). Third, we considered two distinct metrics of profitability, EBITDA and net income. EBITDA (earnings before interest, taxes, depreciation, and amortization) is a measure of the profit from core operations before taxes, without consideration of accounting gains and losses. Net income, often referred to as the "bottom line" or earnings, represents the income or loss considering all revenues and all expenses. This number is used in calculating a company's "earnings per share," which is the most often-quoted measure of a company's profitability.

Annual revenue for BIOTECH and CONTROL is shown in Fig 3A. Except for 1997, when there were only 13 companies in the sample, the revenue for BIOTECH were lower than CONTROL (p<0.0001) for each year in the study period. Median annual revenue for BIOTECH from 1997–2016 was $10.1M vs $192M for CONTROL (p<0.0001).

BIOTECH had higher R&D expenses than CONTROL (p<0.001) each year of the study period (Fig 3B). Median annual R&D spending for BIOTECH from 1997–2016 was $32.9M vs $1.6M for CONTROL (p<0.0001).

The majority of BIOTECH companies had negative earnings before interest, taxes, depreciation, and amortization (EBITDA) throughout the study period (Fig 3C). EBITDA is a measure of pre-tax operating profit without consideration of certain non-cash expenses and non-

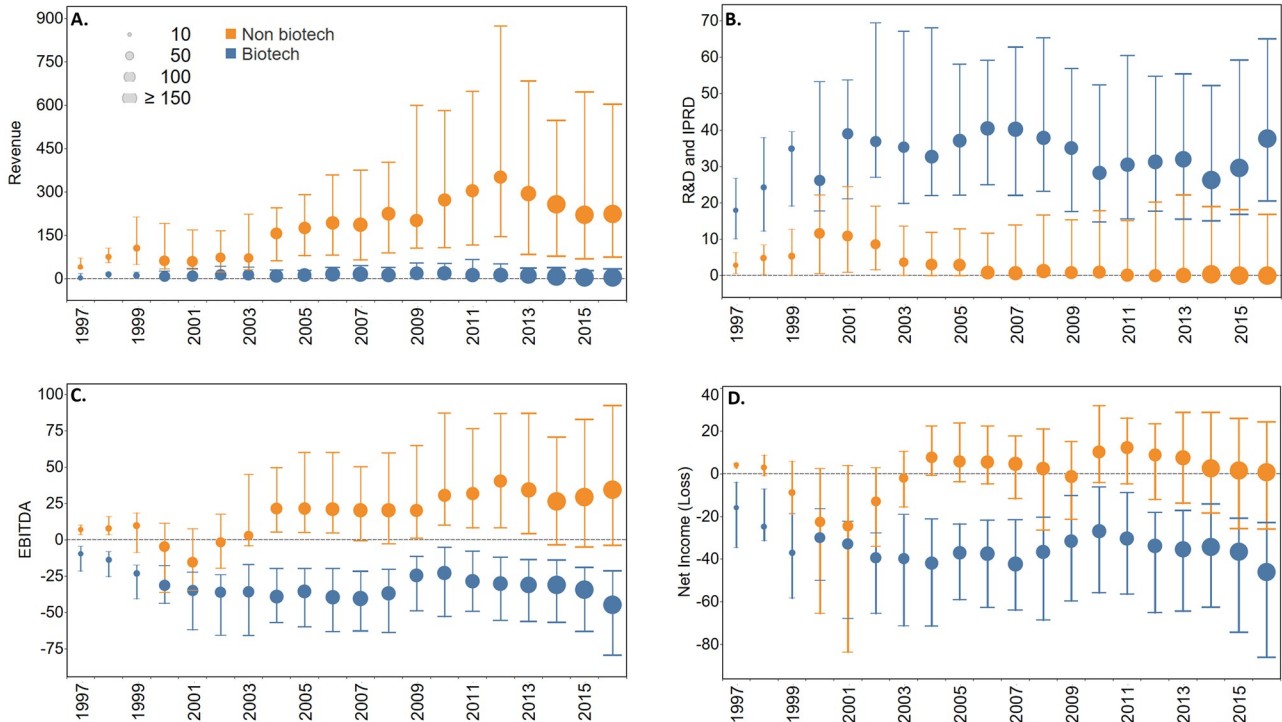

**Fig 3.** A-D. Annual revenue, R&D expense and profit for BIOTECH (blue) and CONTROL (orange) companies 1997–2016. Values shown are the median, 25th percentile and 75th percentile for the year indicated. The number of companies reporting each year is indicated by size of the median marker. A) Annual revenue, B) R&D expense, C) Earnings before Interest: Taxes, Depreciation, Amortization (EBITDA), a measure of operating profit, D) net income or loss, a measure of after-tax profit accounting for all revenue, expenses, gains, and losses, commonly known as earnings. All values in USD, inflation adjusted to 2016.

operating gains and losses, and is similar to Pisano's "operating income before depreciation" [2]. In contrast, while the majority of CONTROL companies had negative EBITDA immediately following the dot.com bubble and through the ensuing recession (2000–2002), they consistently had positive EBITDA from 2003 to the end of the study. Median annual EBITDA for BIOTECH from 1997–2016 was -$33.4M vs $20.5M for CONTROL (p<0.0001).

Similarly, the majority of BIOTECH companies had negative net income throughout the study period, a measure of after-tax profit accounting for all revenue, costs, expenses, and non-operating gains and losses, commonly known as earnings (Fig 3D). In contrast, the majority of CONTROL companies had positive net income each year except during the recession years of 1999–2003 and 2009. The difference in net income was different for each year in the study period except for 2000–2001 (Mann-Whitney p<0.05). Median annual net income for BIOTECH from 1997–2016 was -$36.2M vs $2.86M for CONTROL (p<0.0001).

These data are consistent with the observation by Pisano that very few public companies generated meaningful revenue or profit [1, 2]. For several reasons, however, both our longitudinal analysis and that of Pisano need to be interpreted with caution. First, neither study describes the progression of a consistent cohort of companies and the nature of the cohort exhibits non-random changes over time. For example, during active IPO windows (e.g. 2000, 2013–2016), there are more newly public companies and the average age post-IPO decreases, while during periods of little IPO activity (e.g. 2002–2003, 2007–2009), the average age of the company increases. Second, most of the BIOTECH companies that contributed to development of an approved product were acquired before the end of the study period. This included

46/92 companies that were involved in developing at least one approved product. Thus, most of the revenue from the products developed by the BIOTECH companies would have been accrued after acquisition and would not be recognized as revenue in this analysis.

## Financial outcomes

We examined three distinct measures of financial outcome: market cap, shareholder value, and net value created. Overall, the market cap of BIOTECH grew by $127 billion (104% or 2X), while that of the CONTROL grew by $133 billion (32% or 1.3X). While the market cap of BIOTECH was lower than that of the CONTROL at IPO (Mann-Whitney p = <0.0001), market cap at the last listing of the study period was similar (Mann-Whitney p>0.5) (Table 1).

A majority of companies had a last market cap lower than their first market cap in both the BIOTECH (54%) and CONTROL (55%) datasets (Fig 4A). The proportion of companies with growth versus loss of market cap were similar (z = -0.23, p = 0.81). Considering only the companies that were acquired, the proportion of CONTROL and BIOTECH companies with growth versus loss of market cap was also similar (z = -0.94, p = 0.35).

A total of 134 (42%) BIOTECH and 129 (40%) CONTROL companies achieved a market cap of at least $1B during the study period. Of companies that were acquired, 29 (31%) BIOTECH and 14 (16%) CONTROL companies had a market cap of >$1 billion at acquisition. Conversely, 45 (41%) BIOTECH and 41 (36%) CONTROL that were no longer active at the end of the study period had last market cap <$100 million, indicative of failure and possibly an asset sale.

End shareholder value takes into account both the market cap of the company at the end of the study period, as well as capital returned to shareholders in the form of dividends and stock buybacks. The cumulative change in shareholder value of BIOTECH was lower than that of CONTROL. Overall, the shareholder value of BIOTECH grew by $132 billion (108% or 2.1X) while CONTROL companies grew by $169 billion (40% or 1.4X) (Mann-Whitney p = 0.18). A majority of companies had end shareholder value less than the first market cap in both the BIOTECH (53%) and CONTROL (51%) datasets (Fig 4B). The proportion of companies with growth or loss of shareholder value was similar in both datasets (z = 0.55, p = 0.58). Considering only companies that were acquired during the study period, 51% of BIOTECH and 54% of CONTROLS were below water (z = -0.47, p = 0.64).

Net value created is a measure of the end market cap relative to the net capital raised (total capital raised less capital returned to shareholders in the form of dividends or stock buybacks).

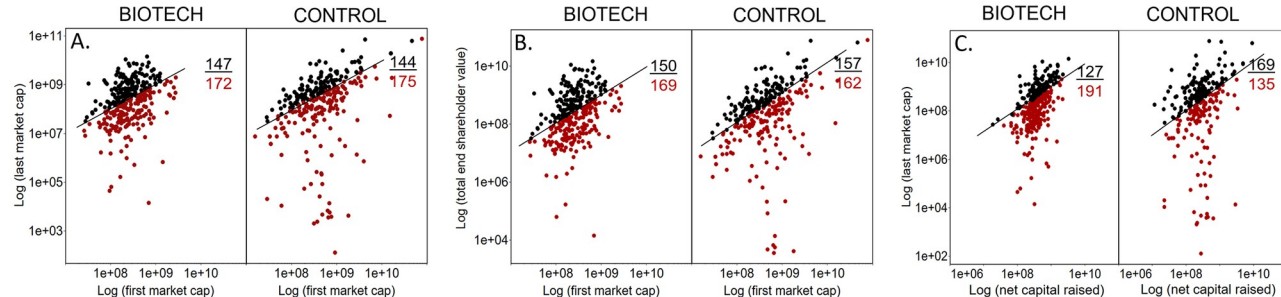

**Fig 4.** A-C. Three measures of value creation by BIOTECH and CONTROL companies 1997–2016. Black markers represent last values greater than first values (value gained), red markers represent last values less than first values (value lost). The number of companies with gains and losses is shown on each panel. A) change in market cap from first market cap (x-axis) to last market cap (y-axis), B) change in total shareholder value from first market cap (x-axis) to end of study, accounting for dividends and stock buybacks (y-axis), C) net value created comparing net capital raised (x-axis) and last market cap (y-axis). All values in USD, inflation adjusted to 2016, shown on a log scale. To see the change in market cap from first to last valuation by IPO window please see an interactive graphic at bit.ly/IPOMKCAP.

BIOTECH companies raised a similar amount of total capital (Mann-Whitney p = .0002) but less net capital (Mann-Whitney p = <0.0001) than CONTROLS. Overall, the net value of BIO-TECH companies increased by $93 billion (53% or 1.6X) while the net value of CONTROLS increased by $411 billion (290% or 3.9X) (Mann-Whitney p = 0.0004).

The majority of BIOTECH companies had an end market cap less than net capital raised (net value loss) (60%). However, the majority of CONTROL (58%) had positive net value, with only 42% having a net value loss. A greater proportion of BIOTECH lost net value (Fig 4C) (z = 4.4, p<0.00001). Considering only companies that were acquired, the same proportion suffered a net value loss (z = 0.33, p = 0.74).

These results are consistent with those of Guo et al, who described a three-year follow-up of 122 biotech IPOs from 1991–2000. He found that, while biotech companies underperformed a paired portfolio by 14.3%, the difference was not statistically significant [13].

## Discussion

Public markets played a critical role in financing emerging biotech companies from 1997–2016, providing $126.8 billion in capital for emerging public companies focused on developing therapeutic products. The continuing public interest in biotechnology investments, which reached record levels in 2014 and again in 2018 [18–20], belies the fact that biotechnology continues to be perceived as a high risk investment.

This work considers the performance of public biotech companies that had IPOs between 1997 and 2016, asking whether these companies performed as well over this interval as a paired CONTROL set of non-biotech companies with concurrent IPO dates. Despite profound differences in the business architecture of BIOTECH and CONTROL companies, these data show that BIOTECH performed as well as the CONTROL on various success metrics including, the length of time they remained listed on public exchanges, the frequency of merger or acquisition, and the fraction of companies achieving market caps of >$1 billion. BIOTECH companies achieved somewhat greater growth in market capitalization and shareholder value, and more BIOTECH companies were acquired with market cap >$1 billion. Overall, 69 BIO-TECHs achieved valuations in excess of $1 billion during the study period. Moreover, BIO-TECH companies were no more likely to lose market cap or shareholder value, enter receivership, or be acquired with valuations of <$100 million, a valuation indicative of failure.

CONTROL companies did substantially outperform BIOTECH in net value created, a measure of end valuation relative to net capital raised. This is consistent with the greater capital requirements of BIOTECH companies in the absence of revenue generating products.

While investments in biotechnology are often considered particularly risky, these data offer little support for that view. The perceived risk of biotechnology is often ascribed to the 90% failure rate and long timelines of product development, and the fact that many companies generate no substantial revenues until products are approved. These data suggest that the risks inherent in biotechnology development may have been mitigated by the emergence of a complementary biotechnology business model.

In a separate study examining the late stage product pipelines of these companies, we show that the companies in this cohort contributed to development of 157 approved products, including 34 first-in-class products and 7 products ranked among the top 200 in sales in 2016 (McNamee et al., in preparation). Significantly, we showed that there was a 52% probability of companies contributing to development of at least one approved product and that the median time from IPO to first approval was 5 years. This estimated probability and timeline for company success is very different than the probabilities and timelines described for any individual product entering development [8] and reflects the fact that most companies in the cohort were

involved in developing multiple products, and that some companies in the cohort had products in late stage development at the time of IPO (McNamee et al., in preparation). This is consistent with the observation by Lo and colleagues that investments in biotechnology "mega-fund" that provided funding for large numbers of products would have a high probability of successfully developing new products and could substantially reduce the risk for investors [21].

The present analysis confirms previous observations that most biotechnology companies have a financial structure characterized by high R&D expense, limited revenue, and negative profits [2]. In fact, only four companies in the BIOTECH cohort had revenues of >$1 billion at the end of the study period (United Therapeutics, Jazz Pharmaceuticals, Biomarin Pharmaceutical, and Horizon Pharma) and only one additional company had >$0.5 billion revenue (Acorda Therapeutics). The paucity of companies generating significant revenue reflects, in part, the fact that the majority of companies that successfully developed products were acquired within several years of their product launch, before product revenue was fully realized. Of the 92 companies associated with at least 1 approved product, only 42 were still active at the end of the study period. Among the others, 15 were acquired before their first product approval, 24 were acquired within five years of their first product approval, and only 7 were acquired more than five years after first product approval.

Together, these data point to the institutionalization of a distinctive business model for public biotechnology companies in which the measure of company success is less likely to be sustained revenues or profits, but rather acquisition. In this model, the risk of investment is mitigated both by pursuing multiple product opportunities and through a portfolio that includes some products in later stage development. In such a model, the valuation of a biotechnology company cannot be meaningfully assessed by the discounted value of future revenues, but must account for the anticipated value of the company at acquisition, which often includes substantial goodwill beyond the estimable value of the company's tangible assets and future product revenues. In fact, one explanation for the similar valuations of BIOTECH and CONTROL companies observed in these data, is that the markets may be benchmarking the value of biotech companies to the value of non-biotechnology companies at a similar stage, whose valuations may be meaningfully calculated by discounting anticipated revenues.

Importantly, these data suggest that the high-risk/high-reward pattern often ascribed to biotechnology investments is not dissimilar to that of paired control companies at a similar stage relative to their IPO. In both datasets, while the majority of companies had negative growth in market cap and shareholder value, and the majority of BIOTECH had negative net value creation, both sectors achieved substantial cumulative value creation by each of these three metrics driven by a small fraction of companies with dramatic gains or losses. Overall, both cohorts exhibited >$100 billion growth in market capitalization and shareholder value, with growth in net value of >$90 billion for BIOTECH and >$400 billion for CONTROLS. Thus, the high risk/high reward pattern is likely a characteristic of public entrepreneurial companies in general, rather than a distinctive characteristic of investments in biotechnology.

There are several limitations to this study. First, these data look exclusively at companies that had IPOs between 1997 and 2016, not the broader biotechnology sector. Specifically, this dataset does not include the first waves of biotechnology IPOs in the 1980s and early 1990s, which produced companies such as Amgen, Biogen, Celgene, Genentech, and Genzyme. Nor does this dataset include the 1991–1994 IPO window, which produced a high number of companies that spawned companies such as Alkermes, Amylin, Cephalon, Gilead, Isis (Ionis), MedImmune, and Vertex (https://www.forbes.com/sites/brucebooth/2012/05/02/biotech-past-biotech-present-reflections-on-the-ipo-window-of-1991-1994/#2c7e6ba65f59). Further research would be required to extend this controlled analysis to the biotechnology sector as a whole.

Second, this dataset contains a disproportionate number of companies from the most recent IPO window (2013–2016), for whom long-term outcomes cannot be assessed. The potential bias introduced by the large number of companies within 5 years of IPO was mitigated, at least in part, by the paired experimental design, which paired each BIOTECH company with a non-biotech (CONTROL) company that had a concurrent IPO. This experimental design would not fully account for market asymmetries that may overvalue or undervalue the biotechnology sector as a whole relative to other sectors for periods of time. Future research, with longer periods of follow-up, will be required to fully assess the economic performance of companies with IPOs in the most recent windows.

## Materials and methods

### Company selection (inclusion/exclusion criteria)

Biotech companies focused on developing therapeutic products with IPOs on NASDAQ between January 1, 1997 to December 31,2016 ("BIOTECH" dataset)were identified in BioCentury's BCIQ database (bciq.biocentury.com/home) using the selection criteria: financing date– 1997 to 2016; Region–North America; Financing type–IPO. The BIOTECH dataset includes companies developing New Chemical Entities (NCEs), biologicals, or reformulations, but not vaccines or diagnostic agents. Companies were excluded that were first public on non-US exchanges or did not file annual reports with the U.S. Securities and Exchange Commission (SEC).

For each BIOTECH company, we identified two non-biotech companies with the closest NASDAQ IPO date (www.nasdaq.com/markets/ipos/), excluding companies with NAICS code 3254 (Pharmaceutical and Medicine Manufacturing) or 999990 (Unclassified Establishments), companies that were first public on non-US exchanges, and those characterized as "special purpose acquisition company." In years with more BIOTECH IPOs than non-biotech IPOs, a CONTROL company could be included in more than one pair. A list of companies is provided in S1 Table.

### Financial data

Annual financial data for fiscal years 1997–2016 was obtained from Compustat (Wharton Research Data Services, wrds.wharton.upenn.edu). Metrics included: Revenue–Total (REVT); Net Income (Loss) (NI); Research and Development Expense (XRD); In-Process R&D Expense (RDIP); Earnings before Interest: Taxes, Depreciation, Amortization (EBITDA); Purchase of Common and Preferred Stock (PRSTKC); Sale of Common and Preferred Stock (SSTK); Dividends–Total (DVT), and Reason for deletion (delisting) (DLRSN), Daily closing stock price (PRCCD) and shares outstanding (CSHOC) were extracted from Compustat Security Daily. IPO data was extracted from Bloomberg Data Services (www.bloomberg.com/professional/solution/data-and-content/) including Offer Price (EQY_INIT_PO_SH_PX) and Shares Offered (EQY_INIT_PO_SH_OFFER) (after greenshoe). Pre-IPO (private) investment was extracted from S-1/A or 424B filings as Total Consideration from existing shareholders.

R&D is calculated XRD + RDIP (RPID is reported as a negative number) to correct for R&D expensed at the course of acquisition. The IPO Offer Size was calculated as Offer price x Shares Offered. Post-IPO investment is equivalent to SSTK. Market cap on the first and last day of trading between January 1, 1997 and December 31, 2016 was calculated as Price Close–Daily x Shares Outstanding (PRCCD x CSHOC). Total capital raised was calculated as pre-IPO investment + IPO offer size + post-IPO stock sales. Net capital raised was calculated as Total Capital Raised–Dividends–Purchase of Common and Preferred Stock. First shareholder

value is equivalent to the first market cap. End shareholder value was calculated as:

$$End\ Market\ cap + Cumulative\ Dividends + Cumulative\ Stock\ Buybacks$$

Net value created was calculated as:

$$Last\ Market\ Cap - Net\ Capital\ Raised$$

All financial data are in USD, adjusted for inflation to 2016 using the Consumer Price Index (CPI-U)-All Urban Consumers data (https://data.bls.gov/cgi-bin/surveymost).

## Statistical analyses

The normality of dataset was assessed by the Kolmogorov-Smirnov test. Since this test rejected the null hypothesis that the financial data was normally distributed ($P < .001$), statistical analysis was performed using tests of median, rather than mean, values. Non-normal data is described as median and inter-quartile ranges (IQR) and compared using the Independent-Samples Mann-Whitney U Test for non-normally distributed data. Time-to-event analysis was performed using Kaplan-Meier survival analysis.

The Kaplan-Meier time-to-event analysis was performed to calculate the estimated probability over time that a publicly-traded BIOTECH or CONTROL company would be delisted, given the fact that many companies were still listed at the end of the study period and the date of any, eventual, delisting is unknown. In Kaplan-Meier analysis, the estimated probability of an event is estimated by the survival function:

$$S(t) = \prod_{i:t_i \leq t} \left(1 - \frac{d_i}{n_i}\right)$$

$d_i$ is the number of events happening at time $t_i$ and $n_i$ is the number of subjects at time $t_i$. In our study, $d_i$ is the number of companies delisted at time $t_i$ and $n_i$ is the total number of companies still trading at time $t_i$. At the end of the study period, companies that have not been delisted are considered "censored" and are not included in $n_i$ at time $t_i$. The estimated probability of a company becoming delisted after its IPO is the multiple of the probabilities at all time intervals.

The proportion of companies gaining/losing value was assessed using the 2 population proportions test (both z-scores and p-values are reported). Because this analysis involved 10 comparisons, a Bonferroni Correction of n = 10 was applied to the p-values to adjust for multiple testing. Statistical analyses were performed using IBM SPSS v26 or Excel. Graphical analysis was conducted in Tableau 2019.4.

## Supporting information

**S1 Fig. Modeling time-to-delisting of BIOTECH (blue) and CONTROL (orange) companies by IPO window using Kaplan-Meier survival analysis.**
(TIF)

**S1 Table.**
(XLSX)

## Acknowledgments

The authors thank Drs. Michael Boss, Nancy Hsiung, and Gregory Vaughan as well as Steven R. Wasserman for contributing expert advice on the analysis.

## Author Contributions

**Conceptualization:** Ekaterina Galkina Cleary, Laura M. McNamee, Fred D. Ledley.

**Data curation:** Ekaterina Galkina Cleary, Laura M. McNamee, Skyler de Boer, Jeremy Holden, Liam Fitzgerald.

**Formal analysis:** Ekaterina Galkina Cleary, Laura M. McNamee, Fred D. Ledley.

**Funding acquisition:** Fred D. Ledley.

**Investigation:** Ekaterina Galkina Cleary, Laura M. McNamee, Fred D. Ledley.

**Methodology:** Ekaterina Galkina Cleary, Laura M. McNamee, Fred D. Ledley.

**Project administration:** Ekaterina Galkina Cleary, Laura M. McNamee, Fred D. Ledley.

**Resources:** Fred D. Ledley.

**Supervision:** Fred D. Ledley.

**Validation:** Fred D. Ledley.

**Visualization:** Ekaterina Galkina Cleary.

**Writing – original draft:** Fred D. Ledley.

**Writing – review & editing:** Ekaterina Galkina Cleary, Laura M. McNamee, Fred D. Ledley.

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
