## [Decision Letter · Decision Letter 0]

16 Sep 2020

PONE-D-20-04037

Comparing long-term value creation after biotech and non-biotech IPOs, 1997-2016

PLOS ONE

Dear Dr. Ledley,

Thank you for submitting your manuscript to PLOS ONE. After careful consideration, we feel that it has merit but does not fully meet PLOS ONE’s publication criteria as it currently stands. Therefore, we invite you to submit a revised version of the manuscript that addresses the points raised during the review process.

First of all, let me please apologize for the delay in sending you this decision. In my opinion, your paper is almost ready for publication. As you may see from your reviewers, both of them consider your paper worthy and very interesting. So do I. As you know, our first reviewer already has accepted your paper for publication, while the other referee is asking you some basic explanations and to clarify some issues related with  with your methodology, that in my opinion you should overcome neatly. 

We look forward to receiving your revised manuscript.

Kind regards,

Alejandro Raul Hernandez Montoya, Ph D

Academic Editor

PLOS ONE

Journal Requirements:

Additional Editor Comments (if provided):

Reviewers' comments:

Reviewer's Responses to Questions

**Comments to the Author**

1. Is the manuscript technically sound, and do the data support the conclusions?

Reviewer #1: Yes

Reviewer #2: Partly

2. Has the statistical analysis been performed appropriately and rigorously? 

Reviewer #1: I Don't Know

Reviewer #2: Yes

3. Have the authors made all data underlying the findings in their manuscript fully available?

Reviewer #1: Yes

Reviewer #2: Yes

4. Is the manuscript presented in an intelligible fashion and written in standard English?

Reviewer #1: Yes

Reviewer #2: Yes

5. Review Comments to the Author

Reviewer #1: As far as I understand the approach the authors too a relevant question is new and very interesting.

On the one hand the fact that pairs are chosen is relevant to this approach and probably increases significance of the approach, but the criteria of the choice, adequately discussed in the article, obviously leave room for arbitrariness. On the other hand the very fact taht pairs with comparable dates and other conditions have been chosen, gives the entire study a strong background of credibility. Yet credibility and statistical significante are not the same, but I personally weigh credibility as a very important factor. This is the basis for my judgement that the data support the conclusions but that I do not know if the statistics have been performed rigorously, yet I think they have been performed appropriately in the sense that the best was done with the data available.

I wish to emphasize that the authors plainly state what they do and how they obtain their conclusions, allowing the reader to draw his own conclusions, if he so wishes!

As for the relevance, I believe that the present crisis enhances the relevance of this paper, as biotech firms play a really important role, and to know that the high risk assigned to biotech even in venture capital circles may well be exaggerated.

Summarizing I wish to say, that I strongly support the publication of this paper in its presnet form, as it is innovative, clearly written and timely!

Reviewer #2: This paper's objective, to compare the financial performance of BIOTECH companies with the performance of NON-BIOTECH companies, is very interesting. However, there are needed the following points: (1) more rigorous explanation on the methodology techniques used within the research (especially on the Kaplan-Meier time-to-event analysis); (2) to provide significance information about the arguments of choosing the financial variables (annual revenue, R&D expense, EBITDA, net income or loss), considering the differences in the business architecture of BIOTECH and NON-BIOTECH companies; and (3) the reasons of the results instead of just analysis. What are the implications of the results? A significant part of the research is only a descriptive analysis. The topic of this manuscript falls within the scope of PLOS ONE.

6. PLOS authors have the option to publish the peer review history of their article (what does this mean?). If published, this will include your full peer review and any attached files.

Reviewer #1: **Yes: **Thomas H. Seligman

Reviewer #2: No

---

## [Author Response · Author response to Decision Letter 0]

2 Oct 2020

Reviewer #1: As far as I understand the approach the authors too a relevant question is new and very interesting.

On the one hand the fact that pairs are chosen is relevant to this approach and probably increases significance of the approach, but the criteria of the choice, adequately discussed in the article, obviously leave room for arbitrariness. On the other hand, the very fact that pairs with comparable dates and other conditions have been chosen, gives the entire study a strong background of credibility. Yet credibility and statistical significant are not the same, but I personally weigh credibility as a very important factor. This is the basis for my judgement that the data support the conclusions but that I do not know if the statistics have been performed rigorously, yet I think they have been performed appropriately in the sense that the best was done with the data available.

Thank you for your comments about the challenge of knowing whether statistical tests are “credible” and meaningful, as well as quantitatively significant. In formal, statistical language, the statistical analysis simply estimates the probability of observing the data we described, assuming the null hypothesis that there is no difference between the BIOTECH and paired CONTROL datasets. In our experimental design, the paired companies were selected to have comparable IPO dates as a control for general market conditions, which is known to be a major factor in any study of company stock performance. 

We would note that treating any particular p value as “significant” is arbitrary. Following the recommendations of the American Statistical Association, this manuscript reports the calculated p-values themselves, and refrains from designating any specific value as significant or not significant. (The ASA Statement on p-Values: Context, Process, and Purpose, https://amstat.tandfonline.com/doi/full/10.1080/00031305.2016.1154108)

I wish to emphasize that the authors plainly state what they do and how they obtain their conclusions, allowing the reader to draw his own conclusions, if he so wishes!

As for the relevance, I believe that the present crisis enhances the relevance of this paper, as biotech firms play a really important role, and to know that the high risk assigned to biotech even in venture capital circles may well be exaggerated.

Summarizing I wish to say, that I strongly support the publication of this paper in its present form, as it is innovative, clearly written and timely!

Reviewer #2: This paper's objective, to compare the financial performance of BIOTECH companies with the performance of NON-BIOTECH companies, is very interesting. However, there are needed the following points: (1) more rigorous explanation on the methodology techniques used within the research (especially on the Kaplan-Meier time-to-event analysis); 

As requested, we have added a section to the Methods section detailing the Kaplan-Meier time-to-event analysis on pg. 19 and added a short, non-technical description of the method in the text on pg. 8. We also added an explanation of the use of the Kolmogorov-Smirnov and Bonferroni correction on pgs. 19 and 20, respectively.

(2) to provide significance information about the arguments of choosing the financial variables (annual revenue, R&D expense, EBITDA, net income or loss), considering the differences in the business architecture of BIOTECH and NON-BIOTECH companies; and 

We have added a section to Results describing each of these variables describing these variables and their relevance on pg. 9. The question about the relevance of accounting metrics given the different architecture of BIOTECH and NON-BIOTECH companies is important. We would note that this analysis focuses explicitly on financial metrics based on data prepared in accordance with U.S. US GAAP standards (Generally Accepted Accounting Practice). These standards are designed explicitly to provide comparability and consistency across industrial sectors and between companies, and are used by investors as benchmarks to guide investments in companies with different architectures. We would not claim that these metrics are perfect comparators, and, as noted in the text, in selecting the paired controls, we eliminated companies in certain areas that provided particularly problematic comparisons.

(3) the reasons of the results instead of just analysis. What are the implications of the results? A significant part of the research is only a descriptive analysis. 

We have extensively revised a portion of the discussion to address the implications of the results, while carefully staying away from any inferences of causality from the statistical analysis. Much of this revision involved consolidating some of the comments originally scattered through the Results section into a more coherent Discussion starting on pg. 14.

---

## [Editor Report · Decision Letter 1]

30 Nov 2020

Comparing long-term value creation after biotech and non-biotech IPOs, 1997-2016

PONE-D-20-04037R1

Dear Dr. Ledley,

We’re pleased to inform you that your manuscript has been judged scientifically suitable for publication and will be formally accepted for publication once it meets all outstanding technical requirements.

Kind regards,

Alejandro Raul Hernandez Montoya, Ph D

Academic Editor

PLOS ONE
---

## [Editor Report · Acceptance letter]

4 Dec 2020

PONE-D-20-04037R1 

Comparing long-term value creation after biotech and non-biotech IPOs, 1997-2016 

Dear Dr. Ledley:

I'm pleased to inform you that your manuscript has been deemed suitable for publication in PLOS ONE. Congratulations! Your manuscript is now with our production department. 

Kind regards, 

on behalf of

Dr. Alejandro Raul Hernandez Montoya 

Academic Editor

PLOS ONE